# Designing PrEP and early HIV treatment interventions for implementation among female sex workers in South Africa: developing and learning from a formative research process

Robyn Eakle,[1,2,3] Nyaradzo Mutanha,[1] Judie Mbogua,[1] Maria Sibanyoni,[1] Adam Bourne,[4] Gabriela Gomez,[1,2] Francois Venter,[1] Helen Rees[1]

[1]Wits Reproductive Health and HIV Institute, Johannesburg, South Africa
[2]Social and Mathematical Epidemiology (SaME), Department of Global Health and Development, London School of Hygiene and Tropical Medicine, London, UK
[3]Sigma Research, Department of Social and Environmental Health Research, London School of Hygiene and Tropical Medicine, London, UK
[4]Australian Research Centre for Sex, Health and Society, La Trobe University, Melbourne, Victoria, Australia

**Correspondence to**
Robyn Eakle;
robyn.eakle@lshtm.ac.uk

## ABSTRACT

**Objectives** The objective of this research was to design relevant, tailored oral pre-exposure prophylaxis (PrEP) and early antiretroviral (ART) interventions for female sex workers (FSWs) in South Africa. This paper examines the methods, process and outcomes of employing an inductive approach to formative research exploring intervention feasibility and acceptability.

**Setting** Research was conducted in several sex work-related settings including five sites in and around clinics and stakeholder offices.

**Participants** Participants in this research included stakeholders, experts in the field and FSWs. This included at least 25 separate engagements, 14 local organisations and 8 focus group discussions (FGDs) with 69 participants, in addition to ad hoc meetings.

**Results** The first set of outcomes consisted of five selected methods: (1) stakeholder consultations; (2) site assessments and selection; (3) field observations and mapping; (4) development of supportive structures to encourage retention and intervention adherence; (5) FGDs conducted with FSWs to explore specifics of acceptability. In terms of feasibility, two sites were selected in central Johannesburg and Pretoria out of five considered. The urban site contexts varied, necessitating adjustments to intervention implementation. There was overall support for PrEP and early ART from stakeholders and FSWs. Concerns included potential issues with adherence to PrEP (and early ART), possible reduction in condom use, resistance to antiretrovirals and burden on scarce resources. These concerns indicated where special attention should be focused on education, messaging and programming as well as development of supportive structures.

**Conclusions** The inductive approach allowed for a wide range of perspectives, defining population needs and accessibility. This research illustrated how similar sex work environments can vary and how implementation of interventions may not be uniform across contexts. Lessons learnt in details could assist in future project designs and implementation of new interventions where feasibility, social and cultural factors affecting acceptability must be considered.

### Strengths and limitations of this study

► This formative research process drew on principles of grounded theory allowing for an inductive, iterative approach to drive the selection of the most appropriate methods for gathering a broad spectrum of data aimed at intervention design.
► Five methods were selected, providing an array of data for decision making and intervention design.
► The final selected project sites were chosen for their nearer uniformity than for diversity; results may not translate beyond this study; however, lessons learnt will be applicable.

## INTRODUCTION

Several guidance documents highlight the need for formative research both when preparing for larger studies and to design the implementation of new public health interventions.[1 2] Formative research includes the assessment of feasibility, reach, acceptability and need of populations to strengthen planned uptake and use of interventions. In particular, formative research aims to ensure the capacity for physical implementation and responsiveness to cultural, social, economic and physical environments.[3–5] However, this phase of work is frequently not reported in detail and important lessons learnt may be lost. This paper describes the detailed decision making and conduct of formative research undertaken to design two new HIV prevention and treatment interventions delivered to female sex workers (FSWs) in a demonstration project in South Africa.

Oral pre-exposure prophylaxis (PrEP) using antiretroviral (ARV) drugs given to HIV-negative individuals to prevent HIV infection has

been shown to be efficacious in multiple clinical trials.[6] In addition, HIV treatment can be given to HIV-positive people as soon as they are diagnosed, called test and treat or early antiretroviral treatment (ART). PrEP and early ART are now recommended in the standard of HIV care by the WHO.[7] These two interventions have also been listed as priorities for operational research. When the study described here was in its infancy, PrEP and early ART were being considered for potential integration into the standard of care in South Africa, which still experiences one of the world's largest HIV epidemics.[8] While South Africa has a generalised epidemic, key populations at highest risk for HIV, such as sex workers, have been in need of prioritised and tailored HIV prevention, treatment and care.[9]

Demonstration projects were recommended by the WHO in 2013 to answer implementation questions relating to feasibility and acceptability of oral PrEP.[10] The call prioritised research for key populations such as sex workers, who have been shown by mathematical modelling to be ideal candidates for PrEP, especially in combination with early ART for HIV-positive people.[11–13] In the previous decade, HIV prevalence among FSWs in South Africa was found to be between 46% and 69%,[14–16] with recent research estimating a prevalence of 72% in the greater Johannesburg area.[17]

Before pursuing implementation research on PrEP and early ART delivery in South Africa, formative research was needed to first shape the interventions. The research presented here employed a comprehensive and inductive approach to formative research that explored feasibility and acceptability of PrEP and early ART, with a view to informing and executing the targeted interventions. This work formed the basis for the design of the TAPS (Treatment And Prevention for Sex workers) Demonstration Project, the purpose of which was to demonstrate how these two interventions could be implemented among FSWs and inform national scale-up in South Africa.[18] We explore and illustrate the approach and process undertaken to define and carry out the formative research, describe how the results informed the overall design of the oral PrEP and early ART interventions for TAPS and reflect on challenges and successes encountered during the process.

## METHODS
This formative research took place between August 2013 and March 2015 and employed an inductive approach based on the principles of grounded theory.[19] An inductive approach to data collection is iterative in nature and, rather than seeking to test a hypothesis or assumption, allows findings to emerge from the bottom up. Findings from one stage of data collection can inform decisions about the selection of subsequent methods and identify new, perhaps unanticipated avenues for data collection to help address the key research questions. Grounded Theory[20] is a qualitative research approach that operates

inductively and can be used to guide robust data collection and analysis in an iterative manner.

Our overall, initial goal was to implement and evaluate oral PrEP and early ART interventions in the TAPS project integrated into a predefined service delivery programme for FSWs. To arrive at this goal, we faced several important questions to address in the formative phase, including:

1. Was there support from stakeholders to test the implementation of PrEP and early ART among FSWs?
2. Where could/should the interventions be delivered, how and by whom?
3. How should FSWs be engaged in the project (both in the formative phase and in the active study)?
4. What structures should be included to support delivery?
5. How did women conceive of acceptability as users of the interventions (and therefore affect demand)?

Since formative research can take many forms including exploring feasibility of supportive structures and logistics for physical delivery of the intervention as well as exploring the acceptability among populations in different contexts,[2] it was important to use an inductive approach to selecting methods driven by the above questions and findings as they emerged. At the beginning of this process, there was a wide original scope within which to consider a large range of logistical possibilities (eg, site locations, adherence support structures, community capacity for involvement) for the design and implementation of the interventions, also an important reason not to predefine all methods which would have limited the scope of design consideration.

Additionally, at the start of the formative work, we knew that the overall aims of the larger TAPS Demonstration Project would be to explore whether FSWs will take up early ART or PrEP, whether the service delivery mechanism was capable of handling the increase in resource needs that might be required, and what the implications of the implementation of these interventions might be, including overall costs should the interventions be scaled up.[18] TAPS was part of the Wits RHI Sex Worker Programme (SWP), a comprehensive health and well-being programme for sex workers running for over 20 years in Johannesburg and other provinces in South Africa.[21] In this regard, the main purpose of the formative research was to develop interventions which had the best possible potential for success as well as for evaluation. This premise drove the formative research process and early on defined the imperative to ensure that sex workers were included throughout.

Decision making about which methods, sites and stakeholders with whom to engage at each step was guided by feedback from community members and the data collected. Principles from the Good Participatory Practice Guidelines developed by UNAIDS and AVAC were also followed, promoting multilevel stakeholder engagement as a core component of research.[1] In line with these guidelines, a range of stakeholders were engaged, including sex workers, sex work-related organisations and

the Department of Health (DoH), further described in the results section.

The selected research activities were chosen according to the evolving data and eventually fell into five core categories of methods: consultations, site assessments, field observations and mapping, development of supportive structures and messaging and FGDs. These methods generated three primary sources of data which informed the design of the interventions: (1) recorded minutes and reports of meetings with stakeholders; (2) notes from field observations collected during engagement with sex worker communities and the environments within which they work as well as the process of hotspot mapping, experiences at potential clinic sites and through the development of supportive structures and (3) transcripts of focus group discussions (FGDs). Outcomes included the chosen formative research methods, the lessons learnt from data collection and the final design of the interventions as well as relevant data collection and monitoring tools and supportive structures. These data were collected by a combination of researchers, sex worker peer educators and clinical staff.

## Data analysis

Data were continuously collected and analysed using an inductive, Grounded Theory approach over 18 months. Field notes, meeting notes, written reports and FGD transcripts were reviewed as activities occurred to identify key themes for further exploration and to define next steps. In this regard, this approach employed a continuous review of data collected in a rolling fashion where the researchers would note significant pivot points in learning to then decide on the next step in research until final decisions on project design aspects had been made. For example, the decision to disqualify a site could be made due to low accessibility of the population only discovered during onsite exploration and then that decision would be immediate and final leading to more attention paid to the next step at other sites. Site selection and staff recruitment represented the end of the first major phase of formative research. Community mobilisation was then focused at the selected sites where testing of messaging, development of supportive systems and development and testing of potential data collection tools continued led by clinic staff, peer educators and potential end users.

FGDs were conducted at each of the final selected sites in multiple languages to suit the participants and analysed following principles of thematic analysis,[22] concentrating on themes originating from the field research incorporated in the FGD guides as well as allowing for organic themes to emerge during the discussions. Further details of the methods and results from the FGDs are presented in a companion paper.[23]

## RESULTS

As this is primarily a methods paper, results are presented in five sections according to the chosen methods as products of the grounded/inductive approach. Sections also relate to the questions addressed by the methods, how the methods were undertaken and lessons learnt influencing the design and execution of the interventions.

These five methods were: (1) stakeholder consultations; (2) site assessments and selection; (3) field observations and mapping; (4) development of supportive structures to encourage retention and intervention adherence; (5) FGDs conducted with potential end-users to explore specifics of acceptability. These are presented in table 1

| Table 1 | Formative research methods and their attributes mapped to research questions | | |
|---|---|---|---|
| **Research question** | **Method(s)** | **Data type** | **Stage/Aspect in process** |
| 1. Was there support from stakeholders to test the implementation of PrEP and early ART among FSWs? | Consultations, Site Assessments, FGDs | Meeting reports and minutes, field notes, FGD transcripts | First step, continuous throughout process |
| 2. Where could/should the interventions be delivered, how and by whom? | Consultations, Site Assessments, Field Observations | Meeting reports and minutes, field notes | Second step, continuous until sites finalised |
| 3. How should FSWs be engaged in the project (both in the formative phase and in the active study)? | Consultations, Field Observations | Meeting reports and minutes, field notes | Incorporated across all steps |
| 4. What structures should be included to support delivery? | Consultations, Site Assessments, Field Observations, Development of supportive structures, FGDs | Meeting reports and minutes, field notes, FGD transcripts | Concluded in later steps of the process, but taken from lessons learnt throughout |
| 5. How did women conceive of acceptability as users of the interventions (and therefore affect potential demand)? | Consultations, Site Assessments, Field Observations, FGDs | Meeting reports and minutes, field notes, FGD transcripts | A component of early phases, but more discretely determined in final steps |

ART, antiretroviral treatment; FGDs, focus group discussions; FSWs, female sex workers; PrEP, pre-exposure prophylaxis.

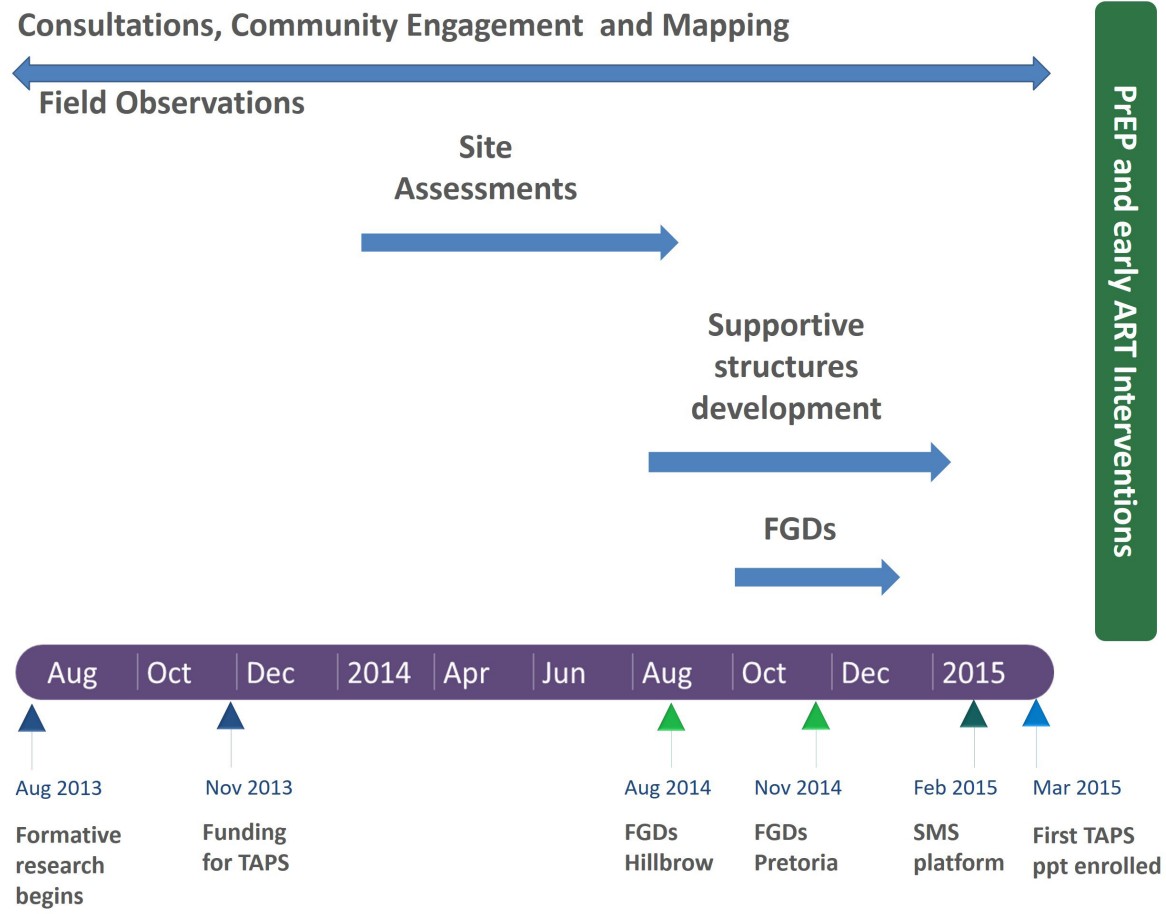

**Figure 1** The formative research process and timeline to design the PrEP and early ART interventions for TAPS. ART, antiretroviral treatment; FGDs, focus group discussions; Ppts, participants; PrEP, pre-exposure prophylaxis; SMS, short message system; TAPS, treatment and prevention for sex workers.

which maps methods to the research questions and data types.

Additionally, a diagram of the methods and the timeline during which they occurred is shown in figure 1. This figure illustrates how some activities were continuous and others discrete, as research is very rarely a clean, linear process and in this case the data were continually informing the decision making.

### Stakeholder consultations

This method sought to answer the question relating to support for the TAPS interventions from stakeholders, but also resulted in addressing some of the other questions around engaging FSWs, potential locations for the research sites and concepts of acceptability. Research began with this activity in order to ascertain a baseline of stakeholder knowledge of and attitudes towards PrEP and early ART as well as attitudes towards sex workers, which denoted initial levels of acceptability for the interventions in this population. We initially conducted

three community consultations in 2013 with a total of 81 attendees from sex worker communities and partner organisations in Hillbrow and Ngodwana, where a new sex worker clinic was in the initiation phase. Additionally, an international consultation of 38 attendees was held in Hillbrow in 2013 and included representatives from sex work communities and organisations, funders, UNAIDS and WHO.[24] These consultations allowed for mapping of organisations and individuals from which to move into the next phase of consultations and formative research.

Smaller meetings were held with local DoH representatives in Johannesburg, Mpumalanga, Phongolo and Pretoria to develop partnerships and potential support agreements for the TAPS project. Updates on, and engagement with, the plans for the TAPS interventions continued at quarterly and other scheduled meetings with local DoH and partners. Parallel meetings were also held with National DoH and the South African National AIDS Council (SANAC). These meetings also informed the site selection in terms of support and local capacity.

Sex workers at each potential site, driven largely by peer educators, helped to identify community organisations to engage for additional perspectives. Stakeholder engagement, both formal and informal, was tracked using an engagement tracking tool[25] developed by the HIV prevention advocacy group, AVAC. Separate meetings were held with 14 different organisations as well as a number of sensitisation trainings provided by the TAPS team and SWEAT to local providers and DoH officials. These added up to at least 25 meetings; however, many additional ad hoc, less formal engagement occurred as well during this time which was not possible to definitively quantify.

Feedback from the consultations pointed to varying degrees of support and acceptability depending on local capacity to take on new interventions and scepticism towards the interventions and particularly PrEP. Stakeholders affirmed that implementing PrEP and early ART together would promote synergistic delivery as well as potentially normalise the use of ARVs in providing options to both HIV-negative and HIV-positive sex workers. However, concerns were voiced about adherence to PrEP (and sometimes early ART) as well as the potential for reduction in condom use, increases in risk of resistance to ARVs and burden on scarce resources. These concerns indicated where special attention should be focused, such as providing evidence about possible resistance to PrEP and strategies for and the philosophy around supporting adherence and condom use, both in early messaging and education for sex workers, implementing and policy partners as well as where to focus messaging and monitoring when implementing the interventions in TAPS.

Finally, consultations were also used to identify potential community advisory board (CAB) members. Meetings were held with individuals expressing interest in participation to inform them about the study and the CAB. These discussions continued until a CAB with up to 12 members (set as the limit for the group which might fluctuate in attendance over time) was established. The CAB incorporated representatives from the local police force, sex worker advocacy organisations, sex workers themselves and religious organisations. A total of five CAB meetings were held before TAPS was launched in March 2015. The CAB became both a result of the formative research and another source of consultation. The CAB also functioned as a supportive structure as members helped spread knowledge of the interventions and the TAPS project as well as lobby for expansion of the interventions to more sites, populations and organisations.

## Site assessments and selection

Site selection primarily addressed the question of where the project could/should be implemented, and much was learnt in this process which is further reported in the field observations. Finalising site selection was a critical step in moving forward with the design of the interventions. We aimed to select 2–3 sites (according to funding and capacity for managerial oversight) with the following population criteria: access to a large number of FSWs (>200 according to early sample size calculations[18]), populations with a relatively balanced proportion of HIV-negative and HIV-positive women and accessibility of clinics. Physical feasibility of the clinics to implement the interventions was assessed through reviews of space, clinical and laboratory infrastructure, required site permissions and approvals and supply chain mechanisms. Support for the interventions was explored with the local DoH and sex worker communities as well as identification of logistical gaps. Site visits, hot spot mapping and site assessments were conducted to determine the feasibility of delivering PrEP and early ART.

Assessments were conducted in four of the original nine Wits RHI SWP sites plus Pretoria as a new site. The other five sites did not meet initial criteria (eg, size of FSW population, clinic feasibility). These five sites featured existing SWP clinics already delivering ARVs per national guidelines or had plans to implement new clinics, interest and support from the local communities and potential access to a large group of FSWs in areas where HIV prevalence was high. This information came from the initial consultations and community engagement and mapping as well as internal programme data. These sites were located in: Ngodwana (rural village in Mpumalanga province), Phongolo (rural trucking site in KwaZulu Natal province), Hillbrow (central, inner-city Johannesburg), City Deep (periurban trucking site immediately south of Johannesburg) and the Pretoria central business district (CBD), the latter three of which were located in Gauteng province. Figure 2 illustrates the sites situated across several provinces in South Africa.

Following the site assessments, three sites were eliminated. Ngodwana was eliminated due to lack of infrastructure and building delays as well as lack of local support. Although there was initial interest, the implementation of the interventions potentially conflicted with limited resources and competing priorities in the area. The rural sex worker community in that location do not self-identify as sex workers which would have created challenges in being able to compare formal sex workers who self-identify and those who do not as part of project evaluation. The prevalence of HIV in Ngodwana was estimated locally to be around 75%–80% in a relatively small village of <300 people mostly made up of sex workers, which also would have made it difficult to implement PrEP.

City Deep was eliminated due to low clinic attendance as well as the highly transient nature of the sex worker populations. Most of the women attending that clinic arrived in the trucks coming from all over South Africa as well as neighbouring countries, which would have made commitment to regular, repeat clinic visits difficult. The Phongolo site, located next to the Swaziland border, was also eliminated as women tended to move in and out of the town and were away for as many as 3–4 months at a time.

As a result, the clinics in Hillbrow and Pretoria were selected as the two TAPS sites. Hillbrow was chosen

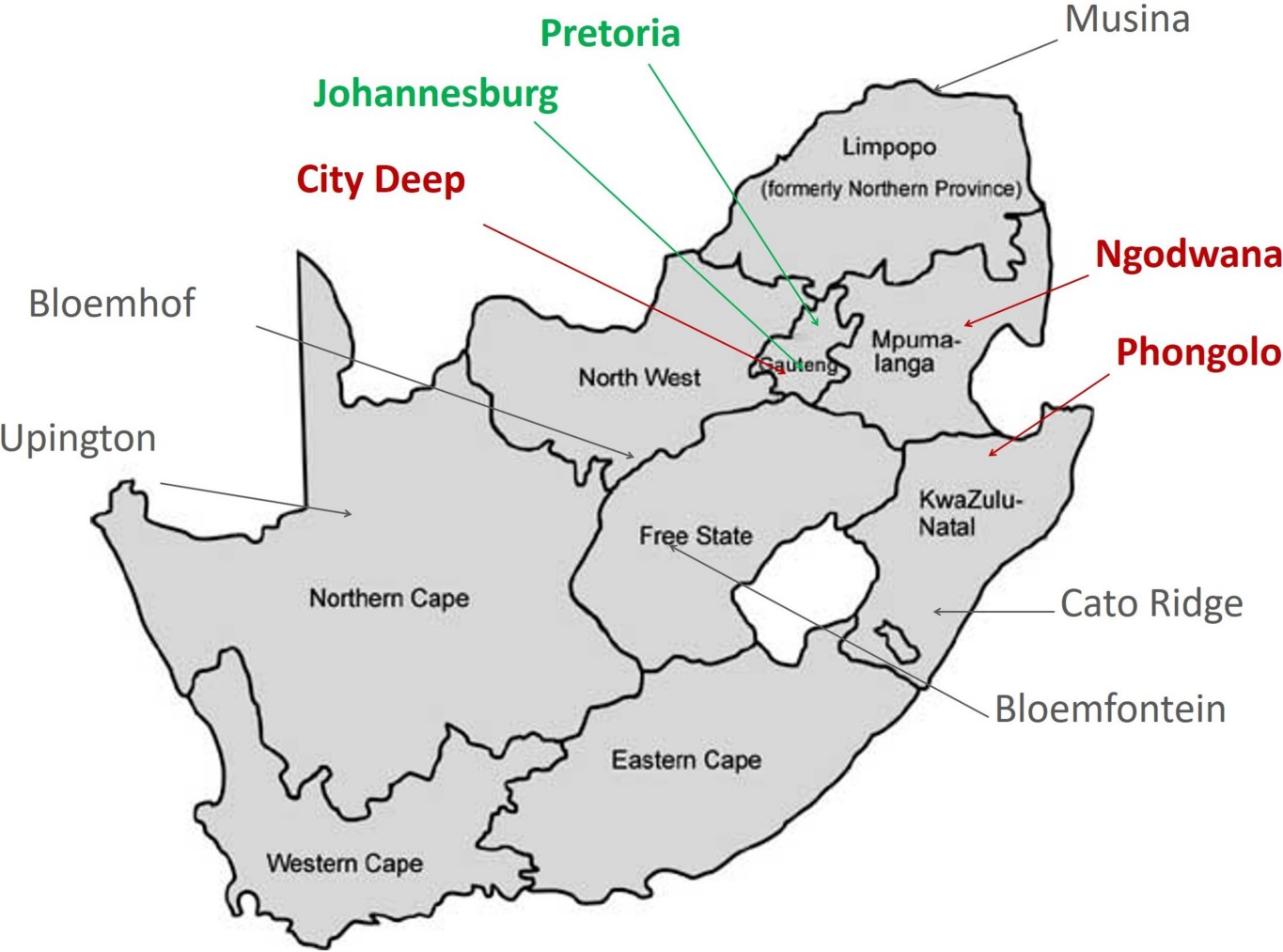

**Figure 2** Map of TAPS and Wits Reproductive Health and HIV Institute (RHI) SWP Sites in South Africa. Note: Green labels and arrows indicate selected sites; Red labels and arrows indicate sites not selected; Grey labels and arrows indicate other Wits RHI SWP sites. SWP, Sex Worker Programme; TAPS, treatment and prevention for sex workers.

because of the long-standing Wits RHI sex worker clinic which maintains strong ties to a large community of sex workers and their managers (brothel owners, 'pimps'). At the time, HIV prevalence was estimated locally at around 50%. The clinic is situated in an area with a high concentration of brothels and street-based locations within walking distance, but also with a number of surrounding outreach areas in the Johannesburg CBD, Yeoville, Jeppe and Rosettenville. For these reasons, this site was considered 'low-hanging fruit' for new intervention implementation, due to the substantial population of women at risk of HIV infection and as a location with solid community support.

In Pretoria, tailored sex worker services and relationships with the community were new at the Sediba Hope clinic in the heart of the CBD; however, there was strong support from the local DoH and expressed need from the sex workers themselves. The newness of such interventions in the area posed a challenge as to whether PrEP and early ART could be implemented within a new sex worker clinic in a new setting and what it would take to do so; however, this was determined to be a potential

strength (and weakness) for implementing these new interventions in a new clinic and the lessons that could be learnt.

### Field observations and mapping

Field observations and mapping elicited information relating to the how of intervention implementation, engaging the community and acceptability. Data were collected through a variety of engagements in the field centring on community mobilisation activities, accruing organically in scope, as discussions and interactions within the communities pointed to additional contacts for engagement. Each point of contact presented opportunities for new consultations and/or locations to conduct outreach. Feedback from the community about potentially offering PrEP and early ART was collected during peer and clinical staff outreach and workshops, which also aimed to elicit values related to acceptability and ultimately generate demand for the interventions.

The core of the community mobilisation and exploration work was conducted through outreach. In this case, outreach consisted of health education, condom

promotion use and a spectrum of services provision from HIV testing and counselling to full clinical services in mobile units in locations accessible to the population.[26] TAPS staff accompanied staff and peer educators from the SWP during their normal outreach activities to probe the local sex worker populations for any knowledge or perceptions of PrEP and early ART as well as start to introduce related information. This resulted in learning more about where and how sex workers operated and how they interacted with health services, informing viability of given sites. We found high interest in and a general cursory understanding of the interventions, particularly in the Johannesburg Hillbrow area where some women had been previously engaged in research studies around PrEP and microbicides. These engagements led us to further think about which supportive structures would be needed and how messages would have to define the differences between PrEP and early ART. Additionally, as core feature of the SWP, interventions were designed around outreach which was identified as a central aspect critical to generating demand and recruiting participants for TAPS.

Hot spot mapping was conducted during outreach to determine the number of FSWs working near the potential sites as well as their working hours which would allow us to focus community education and recruitment efforts. The methods used and built on the results produced by the South African Health Monitoring Study (SAHMS), which drew on multiple mapping methods including time-location and the 'wisdom-of-crowds'.[17] This activity in combination with prior experience derived through the SWP revealed that time and location of sex work is driven largely by client availability. Peak working hours were recorded in each site as critical data dictating when women could be free to attend the clinics. These varied dramatically from site to site, where working hours may begin at 10 am in some places and much later in the afternoon in others. Local languages at each site were also documented so that study materials could be translated and appropriate staff hired.

One of the most important aspects of learning was around the sex work environment, which was somewhat known from our engagement with the larger SWP, but important for staff to see and understand first hand. We assumed, according to previous knowledge, three general categories of sex work locales within the two urban settings. Brothels or hotels usually offer alcohol and entertainment, including combinations of music, dancing and bar games. Many are former or existing hotels while others are simply bars with backrooms. Security protecting sex workers from violent clients is common in these establishments. Second, street-based sex work occurs in high traffic areas where sex workers pick up clients and either join them in vehicles or have sex in a nearby location such as an alley. The third category comprises the 'dark places' (IsiZulu: *mnyamandawo*), informal locations in empty lots or other uninhabited/unoccupied buildings or areas where sex workers build rudimentary structures in which to work.

However, in both cities there is also a fourth, more hidden sex work market found online. Awareness of the online market was generated through conversations with sex workers during outreach and through peer educators, some of whom had online pages and knew of others with similar arrangements. Sex work conducted online usually involves regular clients met through a web-based connection, where business is conducted out of private spaces (personal homes, private spaces run by an owner, upscale hotels or clubs). This population was engaged later in the process as they are less visible to implementers and researchers and difficult to build relationships with given their desire for anonymity.

Finally, earning ability, directly associated with the type of sex work locale, was identified as an important aspect of being able to take time off to come to the clinic even for short periods of time. Sex workers operating in brothels tend to be able to charge higher rates for sex, usually between R50 and R150 (about US$3–10), depending on the status of the brothel and the cost of room rental incurred by the sex workers. On the street, rates are around R50, and in a dark place sex can be R20–30 (about US$1.30–2). Making less money meant more difficulty in taking time away from work to attend the clinic for regular PrEP or treatment related appointments.

### Development of supportive structures and messaging

During consultations, a number of stakeholders provided views on possible components to be incorporated in the intervention design, in particular supportive structures. Supportive structures are generally anything outside of the standard of care which act to promote intervention uptake and use. Mechanisms for supporting intervention adherence and retention in care were explored; however, we decided only to consider those which could be incorporated into a national programme. Among the possibilities were the use of MEMS caps (electronic bottle caps which would count bottle openings as a proxy for pill withdrawals and thus, adherence), pill counts (as conducted in clinical trials) and mHealth solutions in the form of short message service (SMS) messages. After discussing options with DoH partners, it was concluded that SMS, as demonstrated by the ongoing MAMA Connect project,[27] could be scaled up in a national programme, whereas other options would not be feasible.

A participatory process led by representatives of RHI's mHealth and Mental Health teams was undertaken through a series of workshops where themes around healthy living were developed followed by relevant SMS messages by peer educators. Both male and FSW peer educators worked in groups to create the messages which aimed to encourage healthy choices and wellness. These messages took into account the importance of avoiding inadvertent disclosure of HIV status to non-participants and could still be used by the SWP after TAPS concluded. This process produced 110 SMS messages which would be sent once a week in succession to all participants who signed up for the service.

Experiences from early outreach activities directly influenced the messaging employed in each location as well as educational materials used as part of supportive structures for generating demand and promoting awareness of the interventions. Messages focused on defining the interventions, addressing common myths and concerns around side effects and providing information on efficacy and access.

Peer educator and potential user feedback indicated that personal testimonies are highly valued by women in both locations, so these were included on the informational pamphlets. Trained peer educators also enrolled in the study were invited to become ambassadors for the interventions so that they could directly relate their experiences and dispel rumours about side effects.

Finally, additional supportive structures for the interventions and the women using them included the CAB consisting of sex work-related agencies who could help women with legal issues or postrape care who were then also aware of the TAPS project. These helped to build on the existing SWP tried and tested referral systems where we could ensure women would get additional support beyond the scope of the TAPS clinics as needed. One result of the consultations, as well, was the need for holistic sensitivity training around sex work which was conducted with every staff member, from cleaner to clinician, at both sites. Although a more passive support mechanism provided by a local community partner organisation, this was an important, supportive measure in ensuring women felt welcome at the clinics.

## Focus group discussions

The FGDs were the culmination of the formative research exploring acceptability of PrEP and early ART to be delivered in the two clinic sites. Four FGDs were held in each site with a total of 69 participants. The FGDs comprised important final steps in informing design as they explored a more focused community perspective of intervention acceptability on two main levels. One level consisted of data concerning logistics and preferences around physical delivery (location, preferred clinic times, frequency of visits and HIV testing). This also spoke to the 'whom' of service delivery where a significant component of the discussions was the importance of having sensitised nurses and other staff as well as underscoring the best practice of peer-driven education and communication. The other level included social and structural aspects where elements of stigma and socioeconomic norms (eg, where the need for income might supersede health) that might affect uptake and use of PrEP and early ART were explored. Since this paper is focused on feasibility and early stages of acceptability, the end-stage formative data from the FGDs are presented in a companion paper.[23]

## DISCUSSION

In this paper, we have described in detail the formative research process and findings used to inform the design of the PrEP and early ART interventions implemented in TAPS. The grounded, inductive approach afforded a breadth of information around potential site locations as well as community and stakeholder perspectives, including potential opportunities barriers to successful implementation and nuanced aspects of the urban sex work settings. These aspects were critical in considering feasibility of implementation and how to ensure the interventions accomplished reach, accessibility, acceptability and filled the needs of potential end-users. This process allowed for consolidation of a broad scope of lessons learnt to inform intervention design.

Feasibility played a significant role in early decision making as to whether PrEP and early ART could be implemented, and acceptability from stakeholders, including sex workers themselves, ran in parallel. Questions of feasibility addressed site capacity, experience in delivering ARVs, experience with and access to FSWs, site locations related to FSW populations and existing supply chains. Early acceptability was determined through consultations and engagement with potential end users in the community during outreach. Factors influencing feasibility and acceptability would not have been as comprehensively understood, such as the day-to-day clinic operation and the physical contexts and locales of sex work, without repeated site visits and time spent in the field. Continuous, in-depth interactions with the women themselves allowed the TAPS team to better understand sex workers' needs in addition to how interventions reach could be maximised. Additionally, identifying and addressing potential biases from providers and other stakeholders in the provision of the interventions was essential to avoid issues with maintaining permissions and supply chains as well as supportive services.

Generally, it would be expected that sex work populations and industries, with similar urban contexts and in relative geographical proximity, would be the same in both places and that interventions could be implemented uniformly in both locations. However, the formative research demonstrated just the opposite. While there were some similarities in how women live and work in terms of the types of locales and spaces, there were also significant differences. Sex work locales can be similarly categorised in the two cities, but the organisation of these spaces and the make-up of them within Hillbrow and Pretoria is quite different in terms of women's personal safety and earning capability, for example. This translates into different sex worker populations and market dynamics and the need for adaptations to intervention design. Outreach strategies (groups vs one-on-one discussions) and messaging channels (word of mouth, top down through brothel owners and pimps or developing online contacts) were dictated directly by context and the expressed needs of the FSWs.

While many projects have most likely undertaken similar processes in designing interventions for evaluation, taking an explicit inductive approach and the subsequent outcomes have not been reported. For instance,

the Microbicides Development Programme (MDP) has 102 associated references,[28] the majority of which come from the MDP 301 study.[29] A detailed study protocol and several background papers have been published,[30][31–36] no detail on the qualitative aspects contributing to the design-related decision making was included. By presenting this paper, we argue that those details are important for future study, intervention and product implementation design where previous lessons learnt from formative research and methodology can potentially support future researchers and implementers as well as inform research processes and design.

## CONCLUSION

The lessons learnt from this formative research process were directly applied to the design and implementation of the PrEP and early ART interventions delivered through the TAPS project. The inductive approach afforded the opportunity to adapt and include voices and perspectives, which might have otherwise been missed and clarified the needs of the population as well as how to reach them appropriately. This research illustrated how sex work environments can vary, even when the settings are very similar, and therefore implementation of interventions is unlikely to be uniform across contexts. Formative research is critical in designing interventions, especially in new environments but also in well-known contexts. Including intensive stakeholder engagement in formative research will help to ensure that interventions are designed with feasibility and relevance for populations in mind.

## DECLARATIONS

### Ethics approval and consent to participate

Formal ethical approval was provided by the Wits Human Ethics Research Committee (reference number: M131009) for the FGDs which included information testing and community engagement in the field to inform the development of discussion guides. All documents containing data were saved electronically in central folders with limited access to relevant project staff. No participant identifiers were included in any of the reports that were produced. With regard to other formative research elements, no individual data were collected or reported, and ethical principles of the Helsinki Declaration were strictly adhered to as part of this iterative and flexible approach.

### Patient and public involvement

This research was entirely based on potential patients' priorities, experience and preferences. Both potential end-users and other stakeholders were involved throughout each step of the formative research and in the active TAPS project as well, as described throughout this paper. The Wits RHI SWP is based on a sex worker peer-led model of service which supported all of this work (including general outreach as well as recruitment for

the FGDs). Results of this work were disseminated to the community and stakeholders as decisions were made and through the implementation of TAPS.

**Acknowledgements** We would like to thank the many women who participated in this formative research as well as the stakeholders and staff who engaged in helping us design the PrEP and early ART interventions for the TAPS Demonstration Project.

**Contributors** RE developed the research agenda, designed data collection tools, participated in data collection, wrote and collected field notes, analysed the data and drafted the paper. JM participated in data collection and analysed data. NM and MS participated in data collection. AB supported design of data collection tools and data analysis. RE, GG, FV and HR made final decisions on intervention design. All authors reviewed and contributed to the paper.

**Funding** This research was in part supported by funds from AIDS Fonds Netherlands and the Bill and Melinda Gates Foundation (grant OPP1084416).

**Competing interests** None declared.

**Patient consent** Not required.

**Ethics approval** Wits Human Ethics Research Committee.

**Provenance and peer review** Not commissioned; externally peer reviewed.

**Data sharing statement** All data generated or analysed during this study are included in this published article.

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
