## [Reviewer comments · BMJ Open]

ARTICLE DETAILS

TITLE (PROVISIONAL)	Designing PrEP and early HIV treatment interventions for implementation among female sex workers in South Africa: developing and learning from a formative research process
AUTHORS	Eakle, Robyn; Mutanha, Nyaradzo; Mbogua, Judie; Sibanyoni, Maria; Bourne, Adam; Gomez, Gabriela; Venter, WD Francois; Rees, Helen

VERSION 1 – REVIEW

REVIEWER	Frances Cowan LSTM UK
REVIEW RETURNED	18-Sep-2017

GENERAL COMMENTS	This is a useful description of formative research undertaken to inform the TAPS project in Gauteng SA which included a PrEP demonstration project. The work is clearly presented and well written. As stated by the authors it is unusual to find reports of this process outlined with this degree of detail and I think other researchers and programmers (to a lesser degree) will find it useful, although I would take issue with the idea that this was conducted in a timely fashion - very few programmes would have the luxury of this much time to design an intervention and plan for scale up. I was somewhat surprised to see this set out as using an inductive approach as it would have been highly unusual not to undertake the work as outlined prior to setting up a project of this nature. Good participatory practice would have demanded (a priori) that the team spoke to stakeholders, sex workers and other groups as outlined. None the less as I say I think it well presented and useful piece of work
--

REVIEWER	Matt A Price International AIDS Vaccine Initiative, USA
REVIEW RETURNED	20-Sep-2017

GENERAL COMMENTS	Review bmjopen-2017-019292 Thank you for the opportunity to review “Designing PrEP and early HIV treatment interventions for implementation among female sex workers in South Africa: developing and learning from a formative research process” by Eakle et al (bmjopen-2017-019292). This is an interesting paper that could be a very useful tool to help with the design and implementation of service delivery for hard to reach populations in LMICs. However, I feel that the authors could do a better job of
---

concisely explaining the background and rationale for their design, to help public health persons and scientists without a background in grounded theory understand how these results might be replicated elsewhere. Mixing up the methods and results sections was confusing and interfered with getting the message across. In general, I would revisit the paper to reduce the wordiness – specific comments follow.

Abstract: I'm an HIV epidemiologist, and understanding how to improve the design and implementation of PrEP and ART programs for hard to reach populations is of great interest to me. However, from the abstract as written, I wasn't able to entirely understand what you did, why you did it, and what you learned. The "objectives" section talks about design and execution of PrEP and ART programs, however the results section a few lines later just mentions site selection. Your "Results" section reads like methods. Should this instead be "Methods & Results" perhaps? Line 23 seems awkward – results shouldn't consist of "methods chosen" unless I am misunderstanding something about your study? From your methods section, several pages later, this assessment was the initial phase of the TAPS demonstration project, to test how well FSW take up PrEP and ART. I think that's important to mention somewhere in the abstract (perhaps as part of the "setting" section). The results include nothing about the implementation of PrEP and ART programs for FSW, aside perhaps from site selection. It seems there's more results to report – e.g., this process took over 1.5 years, you could note that careful execution of these types of programs is time consuming and teams should be prepared for this? "Inductive approach" is mentioned four times, but not explained. Perhaps imagine how you might write the abstract without using the term "inductive approach" or "grounded theory"? I recommend you revisit your abstract, and give it a significant rewrite.

Introduction: Your paper is of interest to policy implementers, public health officials and scientists like me. Perhaps it's just me, but I'm not familiar with the major concepts that drive this work. Please add several sentences to explain what an "inductive approach" is, and what "grounded theory" is. Consider stating this right away, in the first or second paragraph. Without this, I was unable to understand why these things are helpful in designing your PrEP and ART programs, and why this paper is novel or relevant. (e.g., page 4, lines 23-25 is where I get a first hint of what you mean by these concepts)

Methods: I had a little trouble following your methods. Consider starting your methods at page 3, line 56, and moving your first paragraph on formative research elsewhere – that read like something that might reside in the discussion section. I was unclear about what exactly you did, and in what order. I like how you've laid out your results, why not mirror your methods to follow this? I do realize that as data were collected, it would inform next stage activities, but can you spell out the order of events (i.e., what you show in figure 1, perhaps?) a bit better? E.g., P4, line 31-32: with whom did you have consultations and discussions to inform your methods sites and stakeholders? How many meetings did you have (this might also be shared in the results instead)? Or maybe, with whom did you start, and how did you progress through groups of key stakeholders? In reading your results, I see you add some details on the methods – I think the reader might be better served to find those details in the methods section and not in the results section.

	Results: This section is much longer than it needs to be. Don't tell your reader what you did or why you did it (that's methods), just focus on what you found. How many consultations or FGD, how many individuals, and what did you learn? Minor point but consider numbering your Results subheadings to help the reader navigate this section. P6, line 35: how were these ad hoc data considered or analyzed? P6, line 40: what does "up to 12 members" mean? 1-12? How many CABs were formed? Section on Community Engagement and Mapping: this section is almost all methods. What were the results? What did you learn that helped develop your PrEP/ART intervention outcome? P7, line 28 is all you say, and that just says 'see below'. P7 line 33: why'd you eliminate other sites? Site Assessments and Selections section: your objective is a PrEP/ART intervention. I don't see how this section is relevant to this. At the very least, this could be reduced dramatically, to a short paragraph. e.g.: you need populations at risk in order to implement an intervention of this sort, you assessed X communities, and found that communities A and B were the most in need. I found this section very distracting from your main message. It might make a fine stand-alone paper, but it doesn't feel like it belongs here. P10 line 48: how is the CAB a supportive structure? I didn't understand this sentence. P11 line 16-17: This is the first time I learned your paper was about feasibility and early stages of acceptability! In your methods you say your outcome is "the final design of the intervention" along with the tools to support your intervention (the very next sentence, starting your discussion, confirms this!). This statement is very confusing and you must include at least a high level summary of your FGD findings Your discussion is a very nice summary of your findings – I would take this writing style and apply it to your methods and results, which were more challenging to follow but much more important as it's there you give your details and specifics. What is missing from the discussion is how your work fits into the wider literature. What other work has used similar strategies to develop an intervention like this? What did you do differently, or how did you advance our ways of thinking about developing these types of programs? Figure 1: I had trouble understanding this figure. The arrow-circle surrounding consultations and field observations is confusing. Looks like you're traveling back in time. Is this part of the "inductive approach" whereby field observations influence consultations which then influence future field observations, etc.? I'm not quite sure how to represent this, perhaps placing these two concepts adjacent, with a side-to-side arrow to suggest your inductive approach? You could then show a blue arrow to indicate over what time period these activities were taking place? At what point were the CABs formed? Did they have adequate time to inform your PrEP/ART intervention outcome? I also didn't quite understand how some activities were continuous and some were discrete. A FGD, or a stakeholder consultation, is a discrete event. What's the continuous bit, or what do you mean by continuous?
--	--

VERSION 1 – AUTHOR RESPONSE

Reviewer Name: Frances Cowan
Institution and Country: LSTM, UK
Competing Interests: No competing interests

This is a useful description of formative research undertaken to inform the TAPS project in Gauteng SA which included a PrEP demonstration project. The work is clearly presented and well written. As stated by the authors it is unusual to find reports of this process outlined with this degree of detail and I think other researchers and programmers (to a lesser degree) will find it useful, although I would take issue with the idea that this was conducted in a timely fashion - very few programmes would have the luxury of this much time to design an intervention and plan for scale up.

I was somewhat surprised to see this set out as using an inductive approach as it would have been highly unusual not to undertake the work as outlined prior to setting up a project of this nature. Good participatory practice would have demanded (a priori) that the team spoke to stakeholders, sex workers and other groups as outlined. None the less as I say I think it well presented and useful piece of work.

Thank you for your comments. It is true that it is impossible really to start any work without any prior knowledge of where you are going with it, which is one reason why this is an inductive approach based on the principles of grounded theory. We have also added some clarity to what type of information we started with and how this method was employed which hopefully addresses your concerns. It is also true that we took 18 months to do this work, largely given to us by delayed approvals from the South African Medicines Control Council to conduct research on PrEP. Most of this time, however, was not funded. We leveraged resources from existing infrastructure to conduct most of this research given that TAPS was to be integrated into the existing Sex Worker Programme.

Reviewer 2

Reviewer Name: Matt A Price
Institution and Country: International AIDS Vaccine Initiative, USA
Competing Interests: No competing interests

Thank you for the opportunity to review “Designing PrEP and early HIV treatment interventions for implementation among female sex workers in South Africa: developing and learning from a formative research process” by Eakle et al (bmjopen-2017-019292). This is an interesting paper that could be a very useful tool to help with the design and implementation of service delivery for hard to reach populations in LMICs. However, I feel that the authors could do a better job of concisely explaining the background and rationale for their design, to help public health persons and scientists without a background in grounded theory understand how these results might be replicated elsewhere. Mixing up the methods and results sections was confusing and interfered with getting the message across. In general, I would revisit the paper to reduce the wordiness – specific comments follow.

Abstract: I’m an HIV epidemiologist, and understanding how to improve the design and implementation of PrEP and ART programs for hard to reach populations is of great interest to me. However, from the abstract as written, I wasn’t able to entirely understand what you did, why you did it, and what you learned. The “objectives” section talks about design and execution of PrEP and ART programs, however the results section a few lines later just mentions site selection. Your “Results” section reads like methods. Should this instead be “Methods & Results” perhaps? Line 23 seems awkward – results shouldn’t consist of “methods chosen” unless I am misunderstanding something about your study? From your methods section, several pages later, this assessment was the initial phase of the TAPS demonstration project, to test how well FSW take up PrEP and ART. I think that’s

important to mention somewhere in the abstract (perhaps as part of the “setting” section). The results include nothing about the implementation of PrEP and ART programs for FSW, aside perhaps from site selection. It seems there’s more results to report – e.g., this process took over 1.5 years, you could note that careful execution of these types of programs is time consuming and teams should be prepared for this? “Inductive approach” is mentioned four times, but not explained. Perhaps imagine how you might write the abstract without using the term “inductive approach” or “grounded theory”? I recommend you revisit your abstract, and give it a significant rewrite.

Thank you for this feedback. This is a difficult paper to fit into a traditional abstract format since it is largely a methods paper, and quite different to most methods papers at that. We have revised the abstract to match revisions in the rest of the paper and hope it is clearer now.

Introduction: Your paper is of interest to policy implementers, public health officials and scientists like me. Perhaps it’s just me, but I’m not familiar with the major concepts that drive this work. Please add several sentences to explain what an “inductive approach” is, and what “grounded theory” is. Consider stating this right away, in the first or second paragraph. Without this, I was unable to understand why these things are helpful in designing your PrEP and ART programs, and why this paper is novel or relevant. (e.g., page 4, lines 23-25 is where I get a first hint of what you mean by these concepts)

We have made some significant adjustments for readability and clarity in the Introduction. The additional clarity on the grounded approach has been added in the Methods so as to ensure the Introduction section is not too long.

Methods: I had a little trouble following your methods. Consider starting your methods at page 3, line 56, and moving your first paragraph on formative research elsewhere – that read like something that might reside in the discussion section. I was unclear about what exactly you did, and in what order. I like how you’ve laid out your results, why not mirror your methods to follow this? I do realize that as data were collected, it would inform next stage activities, but can you spell out the order of events (i.e., what you show in figure 1, perhaps?) a bit better? E.g., P4, line 31-32: with whom did you have consultations and discussions to inform your methods sites and stakeholders? How many meetings did you have (this might also be shared in the results instead)? Or maybe, with whom did you start, and how did you progress through groups of key stakeholders? In reading your results, I see you add some details on the methods – I think the reader might be better served to find those details in the methods section and not in the results section.

The Methods section was actually originally laid out like the Results but it was too long. Additionally, most of the Methods were chosen as direct results of learning from other Methods and therefore, as a methods paper, are part of the Results. We have rearranged and re-written the Methods section to hopefully clarify and justify the reasoning behind this work.

Results: This section is much longer than it needs to be. Don’t tell your reader what you did or why you did it (that’s methods), just focus on what you found. How many consultations or FGD, how many individuals, and what did you learn?

This was a challenging paper to write and your comments here reflect some of the back and forth that happened during the writing. We made the decision that since this is a methods paper, and in grounded theory choosing methods can often be part of the results, we wanted to reflect that in the Results section. We have taken your comments on board, however, and have tried to further revise and clarify. As a result, the Introduction, Methods, and Results sections have been rearranged and reframed which hopefully flow more logically now.

Minor point but consider numbering your Results subheadings to help the reader navigate this section.

We agree that this is somewhat confusing, but the clarity for the differentiating headings should come through when the journal formats the manuscript for publication.

P6, line 35: how were these ad hoc data considered or analyzed?

This is explained in the Data Analysis portion of the Methods section.

P6, line 40: what does “up to 12 members” mean? 1-12? How many CABs were formed?

There was only one CAB formed. Often these groups fluctuate in numbers from meeting to meeting according to members' availabilities so there was a cap of 12 members on the group. We have clarified this in the text.

Section on Community Engagement and Mapping: this section is almost all methods. What were the results? What did you learn that helped develop your PrEP/ART intervention outcome?

We have rearranged the Results section and added some clarity in response to this question on what was learned.

P7, line 28 is all you say, and that just says 'see below'.

We weren't sure what this referred to, but hopefully with the reorganized Results section this is now addressed.

P7 line 33: why'd you eliminate other sites?

This is described in detail in the text in the four paragraphs starting with the last paragraph on page 7.

Site Assessments and Selections section: your objective is a PrEP/ART intervention. I don't see how this section is relevant to this. At the very least, this could be reduced dramatically, to a short paragraph. e.g.: you need populations at risk in order to implement an intervention of this sort, you assessed X communities, and found that communities A and B were the most in need. I found this section very distracting from your main message. It might make a fine stand-alone paper, but it doesn't feel like it belongs here.

This is actually a very central component to the design of the project. Settings, both clinical and sex work, vary dramatically and it was important to show the thinking and background that went into choosing sites as those elements directly influence how the intervention can be delivered. This is because PrEP delivery will be very different in different settings. We agree, however, that this section was too long. Several paragraphs were deleted, and others reorganized into other sections of the Results to improve the flow.

P10 line 48: how is the CAB a supportive structure? I didn't understand this sentence.

The CAB's main function is to support the participants in the study by ensuring the study is attending to their needs and reaching people adequately and relevantly. Part of the engagement of community members before the study started was to get their input on the design, and to create a pool of

potential candidates to form the CAB. It was a fluid process where the engagement then resulted in a CAB which was supportive during the active study period.

P11 line 16-17: This is the first time I learned your paper was about feasibility and early stages of acceptability! In your methods you say your outcome is “the final design of the intervention” along with the tools to support your intervention (the very next sentence, starting your discussion, confirms this!). This statement is very confusing and you must include at least a high level summary of your FGD findings.

Thank you for this comment which made it clear that although feasibility was mentioned in the second sentence of the original Introduction, it obviously didn’t come through clearly enough. We have added some text to the end of the Introduction as well as earlier in the Methods.

Your discussion is a very nice summary of your findings – I would take this writing style and apply it to your methods and results, which were more challenging to follow but much more important as it’s there you give your details and specifics. What is missing from the discussion is how your work fits into the wider literature. What other work has used similar strategies to develop an intervention like this? What did you do differently, or how did you advance our ways of thinking about developing these types of programs?

While other studies may actually conduct their formative research in this fashion, to our knowledge, there are no other papers with this kind of detail in them which makes it difficult to situate this paper in the existing literature. There are many papers on other FGDs or surveys conducted before studies, but those only cover a small piece of the larger formative research picture. We have added a small paragraph about this in the Discussion.

Figure 1: I had trouble understanding this figure. The arrow-circle surrounding consultations and field observations is confusing. Looks like you’re traveling back in time. Is this part of the “inductive approach” whereby field observations influence consultations which then influence future field observations, etc.? I’m not quite sure how to represent this, perhaps placing these two concepts adjacent, with a side-to-side arrow to suggest your inductive approach? You could then show a blue arrow to indicate over what time period these activities were taking place? At what point were the CABs formed? Did they have adequate time to inform your PrEP/ART intervention outcome? I also didn’t quite understand how some activities were continuous and some were discrete. A FGD, or a stakeholder consultation, is a discrete event. What’s the continuous bit, or what do you mean by continuous?

We have revised the figure by flipping the timeline to the x axis and removing the circular arrows. Hopefully the text is clearer now and the text and figure reflect each other better now, but the continuous or repetitive aspect is important because the consultations were ongoing throughout the entire process, some formal some informal, and often repeated with the same people to get consensus and support. We hope this new version of the figure is much clearer.

VERSION 2 – REVIEW

REVIEWER	Matt Price International AIDS Vaccine Initiative, USA
REVIEW RETURNED	14-Mar-2018
GENERAL COMMENTS	The revision addresses my concerns, thank you